# Epidemiology and Integrative Taxonomy of Helminths of Invasive Wild Boars, Brazil

**DOI:** 10.3390/pathogens12020175

**Published:** 2023-01-23

**Authors:** Patricia Parreira Perin, Ivan Moura Lapera, Carmen Andrea Arias-Pacheco, Talita Oliveira Mendonça, Wilson Junior Oliveira, Andressa de Souza Pollo, Carolina dos Santos Silva, José Hairton Tebaldi, Bruna da Silva, Estevam Guilherme Lux-Hoppe

**Affiliations:** Parasitic Diseases Laboratory (LabEPar), Departament of Pathology, Reproduction, and One Health (DPRSU), School of Agricultural and Veterinary Studies (FCAV), São Paulo State University—UNESP, Jaboticabal 14884-900, SP, Brazil

**Keywords:** Acantocephala, Nematoda, wild boar, invasive species, suidae, parasitic diversity

## Abstract

Wild boars (*Sus scrofa*) are a significant invasive species in Brazil. We evaluated the helminth diversity of 96 wild boars in São Paulo state. Helminth infection descriptors were calculated, the species were identified and their 18S, 28S rDNA and internal transcribed spacer (ITS) regions were amplified for phylogenetic analyses. *Ascarops strongylina*, *Strongyloides ransomi*, *Globocephalus urosubulatus*, *Oesophagostomum dentatum*, *Trichuris suis*, *Metastrongylus salmi*, *Metastrongylus pudendotecus*, *Ascaris suum* and *Stephanurus dentatus* and *Macracanthorhynchus hirudinaceus* were identified. *Globocephalus urosubulatus* had the highest prevalence and mean abundance, and most animals had mixed infections with three parasite species. There was no association between parasite intensity and prevalence and host sex and body condition index (*p* > 0.05). Novel DNA sequences were obtained from *G. urosubulatus*, *A. strongylina,* and *S. dentatus*. This is the first study on the helmint diversity of non-captive wild boars in Brazil, and the first report of the occurrence of *M. hirudinaceus*, *G. urosubulatus* and *S. dentatus* in Brazilian wild boars. Non-captive wild boars of São Paulo State did not act as capture hosts for native helminth species but maintained their typical parasites, common to domestic pigs. They may act as parasite dispersers for low-tech subsistence pig farming and for native Tayassuidae.

## 1. Introduction

Invasive species are those introduced by humans to a region, intentionally or not, that establish a population that has expanded its geographic range beyond its original introductory location [1]. Biological invasions are related to ecological, economic, sanitary, and social impacts and are one of the primary drivers of native species endangerment [2]. Despite the efforts to reduce the occurrence of new biological invasions, the rates of invasive species detection show no signs of decrease [3].

Wild boars (*Sus scrofa*, Linnaeus 1758) are native to the Paleartic region and were one of the first animals intentionally introduced worldwide for domestication, hunting and commercial breeding. Due to their high prolificity, ecological plasticity and low predator abundance, wild boars were able to successfully establish in many different Biomes [4], resulting in their classification as one of the most harmful invasive species [5,6].

Several pathogens relevant to both human and animal health were diagnosed in wild boars where they were introduced [7,8,9]. In 2013, the Brazilian environmental agency authorized the hunt of wild boars to control the species’ spread [10], which became the most popular control strategy in the subsequent years in the country [11]. With the popularization of wild boar hunting, the consumption of their meat became a common practice among hunters in Brazil [12].

Some studies have evaluated the helminth fauna of captive wild boars [13,14,15,16,17], but there are no studies on wild populations from Brazil. Knowing that there is a growing need to understand the implications of this invasive species and their parasites on the balance of ecosystems as well as on human and animal health, our study documents the occurrence of helminths in wild boars from São Paulo State, one of the most affected regions of Brazil [18].

## 2. Results

The study was conducted using samples from two different groups of animals, G1 and G2; further details about the groups can be seen in Section 4, “Materials and Methods”. The samples from G1 were used for epidemiological assessment and those from G2 for an integrative taxonomy study.

### 2.1. Epidemiological Analysis (G1)

A total of 13,262 helminths were collected from G1, and the descriptors of helminth infection can be seen in Appendix A. From the ten species found, nine represent the Phylum Nematoda and one the Phylum Acanthocephala. Total intensity range was 82–959 and total mean intensity was 378.9 specimens per host. The four most abundant helminth species were *Globocephalus urosubulatus*, with 7542 (56.9%); *Strongyloides ransomi*, with 4140 (31.2%); *Metastrongylus salmi*, with 906 (6.8%); and *Stephanurus dentatus*, with 628 (4.7%) specimens collected. The following species had considerably smaller total abundance: *Ascarops strongylina*, 20; *Metastrongylus pudendotecus*, 9; *Trichuris suis*, 8; and *Macracanthorhynchus hirudinaceus*, 7 specimens collected; and *Ascaris suum* and *Oesophagostomum dentatum*, with 1 specimen each. *Globocephalus urosubulatus* had the highest prevalence (94.3%) and mean abundance (215.5), whereas *Ascaris suum* and *Oesophagostomum dentatum* were the least-frequent species, with prevalences of 2.9% and mean abundances of 0.03. Most studied animals had mixed infections with three parasite species (42.8%), followed by infections with four (37.1%), five (14.3%), six (2.8%), and two species (2.8%), with a mean of 3.7 parasite species per host. Despite the variety of parasites observed, all animals evaluated had good body conditions. There was no influence of parasite intensity on host BCI, and host sex on the descriptors of prevalence and mean intensity of the parasites (*p* > 0.05). Since most of the hunted animals were piglets, the samples had an age bias, hindering statistical analysis regarding host age.

Four eggs and one oocyst morphotype were identified by fecal examination (Appendix A): Strongylidae, *Trichuris suis*, Metastrongylidae, *Strongyloides* sp. and *Eimeria* spp. The prevalence of parasitism in the stool samples was 86% (19/22). *Stephanurus dentatus* eggs were present in urine samples of the affected animals.

### 2.2. Integrative Taxonomic Analysis (G2)

Seven species of helminths were collected from G2, from which six belonged to the Phylum Nematoda: *Ascarops strongylina, Strongyloides ransomi*, *Globocephalus urosubulatus*, *Oesophagostomum dentatum*, *Trichuris suis*, and *Stephanurus dentatus*; and one belonged to the Phylum Acantocephala: *Macracanthorhynchus hirudinaceus*. The morphometric analysis was based on 10 mature specimens of each sex, except for *M. hirudinaceus*, for which only three adult females and four immature specimens were recovered, and *S. ransomi*, whose parasitic form is composed only of parthenogenetic females.

#### 2.2.1. Morphology analysis

##### *Ascarops strongylina* Rudolphi, 1819

Slender, fusiform, reddish-brown nematodes in vivo. The females are larger than the males and the anterior end is similar for both sexes. It has transverse cuticular annulations and one narrow lateral ala on its left side, slightly posterior to left cervical papilla. The mouth opening is round, with two underdeveloped trilobed lips. The pharynx is straight and made up of spiral rings. The esophagus is claviform and divided into two portions: the muscular portion, anterior and short, and the glandular portion, posterior and elongated. It has two asymmetrical cervical papillae in between which the nerve ring is located. The excretory pore is located slightly posterior to the nerve ring and anterior to the right cervical papillae. 

Males (n = 10): Total length 13.4 ± 0.73, body width at the junction of esophagus to intestine 0.336 ± 0.005, pharynx length 0.10 ± 0.01, esophagus length 3.65 ± 0.12, distance from excretory pore and nerve ring to anterior end is 0.295 ± 0.023 and 0.256 ± 0.012, respectively. There is no bursa copulatrix. The posterior end is ventrally curved and provided with two asymmetrical alae with transverse ridges (the largest caudal ala is 0.33 ± 0.093 wide). The spicules are dissimilar in length and shape. The right spicule is longer and thinner, 2.940 ± 0.151 long, and the left spicule is shorter and wider, 0.515 ± 0.117 long. The cloaca has cuticular ornamentation with a serrated margin and its distance to the posterior end is 0.25 ± 0.037. The gubernacle is situated around the cloaca and resembles an elongated triangular membrane, 0.071 ± 0.08 of length and 0.054 ± 0.01 of width. Surrounding the cloaca are several pairs of genital papillae, with four preanal and one postanal pair of pedunculate papillae and five pairs of sessile papillae. 

Females (n = 10): Total length 17.945 ± 1.09, body width at junction of esophagus to intestine 0.375 ± 0.033, pharynx length 0.11 ± 0.008, esophagus length 3.46 ± 0.48, distance from excretory pore and nerve ring to anterior end 0.417 ± 0.011 and 0.288 ± 0.03, respectively. The vulva (8.525 ± 0.60 in distance to the anterior end) is situated in the middle portion of the body. The ovejector is tubular and contains embryonated eggs (0.04 ± 0.002 long and 0.02 ± 0.001 wide). The tail is short, with a rounded tip. The distance from the anus to the posterior end is 0.305 ± 0.017. 

The morphometric data of *A. strongylina* in this study were compared to those of Dakova and Panayotova-Pencheva [19], and presented in Appendix A. Photomicrographs of taxonomic interest are presented in Appendix A. 

##### *Strongyloides ransomi* Schwartz and Alicata, 1930

Females (n = 10): Only parthenogenetic females are parasitic. They have an elongated, filiform body. The total length is 5.26 ± 0.276, the body width at the junction of the esophagus and intestine is 0.09 ± 0.008, the length of the esophagus is 1.015 ± 0.026, and the distance from the nerve ring to the anterior end is 0.25 ± 0.015. The body width increases gradually from the anterior end to the base of the esophagus. From the ovarian region to the posterior end, the body width gradually decreases ending in a conical, pointed tail. The ovary has loops in the anterior portion that decrease in number towards the posterior end of the body. The ovejector contains thin-shelled ellipsoidal embryonated eggs (0.055 ± 0.005 long and 0.032 ± 0.002 wide). The vulva is a transverse slit of protruding labia located shortly after the middle portion of the body, at 3.26 ± 0.159 from the anterior end. The anus is located at 0.08 ± 0.005 from the posterior end. 

The morphometric data of *S. ransomi* in this study were compared to those of Alicata [20] and Giang et al. [21] and presented in Appendix A. Photomicrographs of taxonomic interest are presented in Appendix A.

##### *Globocephalus urosubulatus* Alessandrini, 1909 

Thin bodied, fusiform, yellowish in vivo nematodes. The females are larger than the males and the anterior end is similar for both sexes. It has a thick cuticle with transverse striations. The oral opening is circular and surrounded by a cuticular ring that forms a buccal capsule containing a pair of teeth near its base. The nerve ring is located slightly anterior to the well-developed and symmetric cervical papillae. The esophagus is claviform and well developed. 

Males (n = 10): Total length 6.17 ± 0.471, body width at junction of esophagus to intestine 0.35 ± 0.023, buccal capsule 0.15 ± 0.014 long by 0.13 ± 0.023 wide, esophagus length 0.62 ± 0.026. The distance from the excretory pore, nerve ring, and cervical papillae to the anterior end is 0.49 ± 0.04, 0.48 ± 0.039, and 0.56 ± 0.042, respectively. The spicules are similar in length and shape, being long and filiform (0.61 ± 0.09 long). The gubernacle is slender and half-moon shaped (0.08 ± 0.001 long). The bursa copulatrix is well developed and wide. Five rays emerge from the dorsal trunk, bifurcate around two-thirds of its length, and each subray trifurcate at its end. The ventrolateral and lateroventral rays are bifurcated, the anterolateral, mediolateral, and posterolateral rays emerge from the common trunk base then split. The externodorsal rays also emerge from the common trunk base in parallel to the edge of the bursa. 

Females (n = 10): Total length 8.4 ± 0.27, body width at junction of esophagus to intestine 0.47 ± 0.054, buccal capsule 0.21 ± 0.017 long by 0.14 ± 0.02 wide, esophagus length 0.84 ± 0.11. The distance of the excretory pore, nerve ring, and cervical papillae to the anterior end is 0.65 ± 0.02, 0.63 ± 0.035 and 0.71 ± 0.017, respectively. The distance from the vulva to the posterior end is 6.67± 0.023. The ovejector contains ellipsoidal morulated eggs with thin shells (0.06 ± 0.002 long and 0.03 ± 0.001 wide). The anal upper lip is prominent and the distance from the anus to the posterior end is 0.22 ± 0.034. The tail is conical.

The morphometric data of *G. urosubulatus* in this study were compared to those of Francis [22] (1978), Nanev et al. [23] (2007), and Pinheiro et al. [24] (2021) and presented in Appendix A. Photomicrographs of taxonomic interest are presented in Appendix A.

##### *Oesophagostomum dentatum* Rudolphi, 1803

Thin-bodied, fusiform nematode with a yellowish coloration in vivo. The females are larger than the males and the anterior end is similar for both sexes. Surrounding the oral opening there is a relatively small buccal capsule that has an outer crown of sharp leaf-shaped lamellae, and a poorly developed inner crown with numerous small lamellae (outer and inner corona radiata). After the buccal capsule begins the esophagus, which has a bulbar dilation in its posterior region. 

Males (n = 10): Total length 8.98 ± 0.038, body width at the junction of the esophagus and intestine 0.21 ± 0.018, length 0.43 ± 0.017 and width 0.13 ± 0.013 of the esophagus, distance from the excretory pore and nerve ring to the anterior end 0.34 ± 0.023 and 0.24 ± 0.008, respectively. Well-developed bursa copulatrix. From a broad common trunk arise the externodorsal rays which do not reach the edge of the bursa. Then from the trunk arises the dorsal ray, which subsequently divides into two branches which subdivide into a shorter lateral branch and a longer medial branch, and only the latter reaches the edge of the bursa. The lateral rays arise from a common trunk, then the anterolateral separate from it, ending before the other lateral rays. The other two lateral rays differentiate after the anterolateral ray but stay adhered to each other until their posterior end. The mediolateral ray is slightly shorter than the posterolateral ray. The ventral rays are symmetric, arise from a common trunk, and stay adhered after differentiation. The spicules are similar in length (0.98 ± 0.118 long) and shape, the anterior end being thicker and the posterior end thinner. The gubernaculum is 0.10 ± 0.007 long and has a sword-like shape directed towards the posterior end. 

Females (n = 10): Total length 12.88 ± 0.409, body width at the junction of the esophagus to the intestine 0.24 ± 0.008, length 0.45 ± 0.02 and width 0.12 ± 0.008 of the esophagus, distance from the excretory pore and nerve ring to the anterior end 0.345 ± 0.021 and 0.24 ± 0.023, respectively. The distance from the vulva to the posterior end is 0.63 ± 0.113. The ovejector contains ellipsoidal eggs with thin shells and morulated embryos (0.06 ± 0.005 long and 0.03 ± 0.001 wide). The tail is shaped like an extended cone and the distance from the anus to the posterior end is 0.27 ± 0.043.

The morphometric data of *O. dentatum* in this study were compared to those of Dakova and Panayotova-Pencheva [19] (2017) and presented in Appendix A. Photomicrographs of taxonomic interest are presented in Appendix A.

##### *Trichuris suis* Schrank, 1788

Slender, whip-like nematode with yellow to brown coloration in vivo. The females are larger than males and the anterior end is similar for both sexes. The anterior region of the body is long and filiform. The body width increases at the junction of the esophagus to the intestine, which remains constant until the coiled posterior end. The males’ posterior end is more tightly coiled than the females. The mouth opening is simple, with no lips or buccal capsule. The esophagus is thin, tubular, and surrounded by glandular stichocytes (stichosome). 

Males (n = 10): Total length 36.784 ± 3.854, esophagus length 2.346 ± 0.29, width of anterior region 0.184 ± 0.07, width at the junction of esophagus to intestine 0.379 ± 0.08, width of posterior region 0.725 ± 0.098. The bursa copulatrix is absent, presence of a single spicule 2.37 ± 0.182 long, covered by an eversible rough spiked sheath with 0.29 ± 0.132 long. Presence of two pericloacal papillae. 

Females (n = 10): Total length 44.77 ± 3.8, esophagus length 3.268 ± 0.201, width of anterior region 0.21 ± 0.04, width at the junction of esophagus to intestine 0.34 ± 0.08, width of posterior region 0.73 ± 0.11. No vulvar prolapses were observed and the vulvar region was covered by cuticular spikes. The ovejector contained ellipsoidal to barrel-shaped yellow-brown eggs with two distinct opercules (0.06 ± 0.005 long and 0.03 ± 0.003 wide). The distance from the vulva to the junction of the esophagus to the intestine is 0.25 ± 0.05, terminal anus. 

The morphometric data of *T. suis* in this study were compared to those of Cutillas et al. [25] (2009) and Nissen et al. [26] (2012) and presented in Appendix A. Photomicrographs of taxonomic interest are presented in Appendix A.

##### *Stephanurus dentatus* Diesing, 1839 

Stout, fusiform nematode with a light yellowish coloration in vivo and a transversely striated cuticule. The thinness of its integument allows the internal organs to be distinguished and a long intestine with a series of grayish circumvolutions can be seen. The females are larger than the males, and the anterior end is similar in both sexes. The oral opening is surrounded by a cuticular ring that forms a thick buccal capsule with around six teeth near its base. A corona radiata encircles the mouth opening. The esophagus is claviform, narrower in its anterior portion and has a bulbar dilation in its posterior portion. 

Males (n = 10): Total length 23.267 ± 1.931, body width at the junction of the esophagus to the intestine 1.97 ± 0.12, length of the esophagus 1.864 ± 0.27, buccal capsule 0.19 ± 0.001 long by 0.18 ± 0.002 wide, distance from the excretory pore and nerve ring to the anterior end 0.575 ± 0.21 and 0.504 ± 0.088, respectively. The spicules are similar in length (1.079 ± 0.018) and shape, with a thick anterior end and a posterior end shaped like an arrowhead. The gubernacle is 0.237 ± 0.073 long and shaped like a half moon. The bursa copulatrix is poorly developed with short stout rays terminated in rounded wide tips. 

Females (n=10): Total length 29.883 ± 0.276, body width at the junction of esophagus to intestine 1.634 ± 0.015, length of esophagus 1.894 ± 0.51, buccal capsule 0.22 ± 0.003 long by 0.201 ± 0. 002 wide, distance from the excretory pore and nerve ring to the anterior end 0.594 ± 0.134 and 0.552 ± 0.038, respectively. The vulva is situated close to the anus and the ovejector is short; the distance from the vulva to the posterior end is 1.174 ± 0.047. The posterior end narrows suddenly after the anus and ends in a small conical tail. The distance from vulva to anus is 1.41 ± 0.013 and from anus to posterior end is 0.31 ± 0.00. There is a papilla on each side of the anus. 

Photomicrographs of taxonomic interest are present in Appendix A.

##### *Macracanthorhynchus hirudinaceus* (Pallas, 1781)

Females (n = 3): Large and robust acanthocephalan, yellowish in vivo. The three females were 304, 315, and 381 mm long and 6, 8, and 9 mm wide, respectively, tapering gradually toward the posterior end. The proboscis is cylindrical, globular, broader in its anterior portion, and flat at its apex, with six spiral rows of six spikes each. The spikes are embedded in the proboscis in elevated cuticular holes. 

The morphometric data of *M. hirudinaceus* in this study were compared to those of Lisitsyna [27] (2019) and Amin et al. [28] (2021) and presented in Appendix A. Photomicrographs of taxonomic interest are presented in Appendix A.

#### 2.2.2. Phylogenetic Relations

All the seven species analyzed had at least one ribosomal region amplified. Although different sets of primers were tested, it was not possible to obtain amplicons from the three main ribosomal regions for all of them (Table 1). The Bayesian phylogenetic analysis grouped the specimens coherently according to the morphological results. All clades presented high reliability values. The phylogenetic trees for the 18S rDNA, 28S rDNA and ITS regions can be seen in Figure 1, Figure 2 and Figure 3.

*Ascarops strongylina*, as identified by morphogical analysis, was grouped in both 18S and 28S rDNA phylogenetic trees with other sequences corresponding to specimens belonging to the family Spirocercidae, to which it belongs. This is the first molecular record of this species.

*Strongyloides ransomi* had only the 18S rDNA region sequenced; even so, it was possible to confirm the species. The sequence obtained (1590 bp) showed 99.94% of identity with another *S. ransomi* sequence from the database and it was phylogenetically grouped with other sequences corresponding to the *Strongyloides* genus.

*Globocephalus urosubulatus* had the 18S rDNA, 28S rDNA and ITS regions sequenced, and it was phylogenetically grouped in clades formed by members of its order, Strongylida, and/or its family, Ancylostomatidae. This is also the first molecular record of this species.

*Oesophagostomum dentatum* sequences of the 18S rDNA and 28S rDNA regions were grouped in the clades composed of members of Chabertiidae family, in which they are currently classified. Sequences of *O. dentatum* correspondent to the fragments sequenced in this study were not found in the database; thus, for these regions, the species that matched and showed higher identity with the *O. dentatum* sequences was *Oesophagostomum muntiacum*, a nematode that parasitizes the large intestine of *Muntiacus* deer. The ITS region sequence of our study was grouped in the phylogenetic analysis to a correspondent sequence of *O. dentatum* from the database.

The genetic identity of *Trichuris suis* was confirmed in the phylogenetic analysis of 18S rDNA sequences. The 18S rDNA sequence of *T. suis* obained in this study (1780 bp) showed 100% of identity with sequences of its species from the database. 

The species *Stephanurus dentatus* was genetically confirmed by the 18S rDNA and 28S rDNA phylogenetic analysis and for presenting 100% of identity with other sequences of this species from the database. There were no ITS region sequences of this species in the database, so the *S. dentatus* ITS region sequence from this study was grouped with sequences from its order, Strongylida.

The *Macracanthorhynchus hirudinaceus* obtained sequences grouped with sequences of its species present in the database in both 28S rDNA and ITS region phylogenetic analysis. Nonetheless, the sequences of *M. hirudinaceus* obtained in this study presented only 94.41% and 98.55% of identity with those from the database, indicating that the specimen found in this study presents considerable genetic distance from the ones deposited in the genbank. 

## 3. Discussion

This is the first study on the helminthological diversity of non-captive wild boars in Brazil, and the first report of the occurrence of *M. hirudinaceus*, *G. urosubulatus*, and *S. dentatus* in this species in the country; the other helminths have been reported in captive wild boars in Brazil [13,14,15,16,17]. A greater species variety was observed in wild specimens in comparison with captive wild boars from the same region [14]. It appears that invasive wild boars in Brazil retained some of their typical parasites common to domestic pigs and did not act as capture hosts for any species native to the Neotropical region.

All helminth species identified have been reported in domestic pigs [29], and native Tayassuidae [30]. Regardless of helminth presence in domestic cycles, wild boars can act as hosts and reservoirs, keeping them in wild cycles. These parasites may occur with a higher intensity and prevalence due to the greater availability of hosts with the presence of the wild boars, and health authorities should be attentive to the emergence and re-emergence risks of these helminths [31]. This may exert more pressure on native populations that are already rarely documented in the studied biomes, especially *Tayassu pecari*, which is listed as vulnerable in the IUCN red list [32]. Wild boars may also act as dispersers of these parasites to domestic swine, especially in extensive low-tech subsistence farming. Although this is not the present model of pig farming in São Paulo state, wild boars have been recorded in 22 of the 26 Brazilian states, which have varied animal husbandry profiles [18].

Almost all predominant species (*G. urosubulatus*, *S. ransomi*, and *S. dentatus*) have monoxenous biological cycles. The high prevalence and intensity of these parasites may be related to behavioral habits of wild boars that may increase contact with parasites’ infective stages, such as wallowing in humid soils and revolving the soil in search for food [29,33]. Another determining factor for monoxenous parasite dispersion is the increasing host population density. A few Brazilian reports indicate numerous wild populations of these animals [18,34]; thus, increasing the infective larvae chance of reaching new hosts to complete their biological cycle. Feed traps prepared by hunters, agricultural plantations, and other feeding sites that attract wild boars have higher levels of environmental contamination and higher risk of infection by monoxenous parasites [35]. Some studies have shown that the greater availability of food also favors wild boar reproduction and reduces mortality rate, resulting in increased host populations and parasite dissemination [35,36]. Earthworms are part of wild boars’ diet and can also play an important role as paratenic hosts for *S. dentatus* and *A. suum*, increasing the survival period of the infective phases of these monoxenous parasites [4,29,37].

Contrary to the low infection indicators observed in this study, Acanthocephala infection is usually quite prevalent and intense in wild boars, mainly in European countries [38], Iran [31], and Jamaica [39].

The absence of a relationship between host sex and parasitic infection indicators may be related to studied animals age range. Adult males with full reproductive activity have solitary habits and occupy a large territory, but males up to two years old are still young adults and live in small groups that are exposed to the same risk factors [40]. Thus, the absence of significant behavioral differences compared to females of the same age group explains the similar parasitic infection indicators in both sexes [41].

This study provides detailed morphologic and morphometric data of the recovered helminth species; there are a few studies about this topic, such as the reports of Dakova and Panayotova-Pencheva [19] on *A. strongylina* and *O. dentatum* from wild boars in Bulgaria; Giang et al. [21] on *S. ransomi* from pigs in Vietnam; Pinheiro et al. [24] on *G. urosubulatus* from domestic pigs in Brazil; and Lisitsyna [27] and Amin et al. [28] on *M. hirudinaceus* from wild boars in Ukraine. Nevertheless, the current analyses in this regard for most of them are few and outdated; this is especially true for *S. dentatus*, as anatomic studies on this species are scarce and dated [42,43,44]; for *T. suis*, as there is no clear differential criterion from *Trichuris trichiura* and new morphological criteria are needed to differentiate these two species [25]; and for *M. hirudinaceus*, as this species appears to exhibit considerable size variation of taxonomically important structures in relation to geographic region and host [28]. The intraspecific variation in the morphometric characteristics of the various helminth species found in relation to previous studies may be due to differences in methods of preparing and observing the specimens, as well as to peculiarities of the parasite and host populations distributed worldwide [19].

We presented five novel DNA sequences, three from *G. urosubulatus* (18S rDNA, 28S rDNA and ITS regions), two from *A. strongylina* (18S rDNA and 28S rDNA regions), and one from *S. dentatus* (ITS region). This is the first molecular study of *G. urosubulatus* and *A. strongylina*, and the sequences obtained and deposited in the genbank certainly will contribute to future phylogenetic analyses. With the exception of *O. dentatum* and *T. suis*, molecular data of the helminth species analyzed in this study are very limited in the literature and genetic databases. The Bayesian phylogenetic results allowed the confirmation of all helmints species of this study that were previously morphologically identified. The combination of both analysis made the morphological and genetic diversity of *M. hirudinaceus* clear. Our results highlight the importance of obtaining sequences of helminths from different localities in order to allow molecular diversity and populational characterization analysis. 

## 4. Materials and Methods

### 4.1. Ethical and Legal Aspects

This study is part of a research project approved by the “Chico Mendes Institute for Biodiversity Conservation”, with the application in the Biodiversity Authorization and Information System SISBIO #55352-1, #62641-2 and #67577, and the Animal Ethics Committee of FCAV/Unesp, protocols #2465/2017 and #3217/2021. The partner fauna management teams have environmental licenses issued by the Federal Technical Registry of the Brazilian Institute of Environment and Natural Resources (IBAMA) to manage invasive exotic fauna and registration certificates to carry and transport hunting weapons issued by the Brazilian Armed Forces. All procedures adopted in this study are in accordance with international standards.

### 4.2. Study Area

The study was conducted in nine cities within São Paulo State’s northeastern region (Figure 4) and covered an area of approximately 43,000 km^2^. The sum of the study area’s estimated population is 327,000 people [45]. The vegetation cover is a transition between the Cerrado and Atlantic Forest Biomes. The predominant climate is humid subtropical (Köppen’s modified climate classification), with average monthly temperatures of 18 °C and average annual rainfall between 1200 and 1500 mm. The region has four to five months of drought in winter, between May and September, and is at an average altitude of 600 m above sea level [46,47,48]. The regional economy is strongly dependent on agricultural activities such as sugarcane, orange, rubber, soybean, corn, peanut, and tomato cultivation, as well as beef cattle and poultry farming [49,50,51].

### 4.3. Sample Collection

Sampling was carried out without biostatistical criteria since it was dependent on the hunting success of the partner fauna management teams. The study was conducted using samples from two different groups of animals; in total, 96 non-captive wild boars were killed, 35 from September 2016 to July 2017 (first group; further on called G1), and 61 from April 2019 to October 2021 (second group; further on called G2). The samples from the first group were used for epidemiological assessment and the ones from the second group for an integrative taxonomy study of the helminths found. The sex of the animals was identified, and their age was estimated according to dental eruption [52]. The wild boars’ organs were removed from the carcasses in the field, packed in individual plastic bags, identified, stored in isothermal boxes with ice, and immediately sent to the Laboratory of Parasitic Diseases (LabEPar) of the School of Agricultural and Veterinary Studies (FCAV) from São Paulo State University (Unesp), where they were promptly processed. 

### 4.4. Helminthfauna Assessment (G1 and G2)

Kidneys, ureters, bladder, and perirenal fat were dissected and inspected macroscopically for the presence of helminths. The gastrointestinal tract was separated into its anatomical segments (stomach, small intestine, and large intestine) and sectioned longitudinally. The contents and mucosa of the segments were washed under running water over 100 µm wire mesh sieves. The respiratory tract was placed in metallic trays, slit open, thoroughly washed, and sieved in Tyler 100 metallic sieves (G1). The whole content (G1) or aliquots (G2) of the material retained in the sieves and the helminths found during macroscopic examination were placed in bottles containing Railliet and Henry’s solution (G1) or 70% alcohol (G2) for later separation and identification using a Leica EZ4 HD stereomicroscope (Leica Microsystems Inc., Buffalo Grove, IL, USA). The recovered helminths (G2) were then fixed in absolute ethanol (Merk, Darmstadt, Germany) and stored at −20 °C until processed.

#### 4.4.1. Morphological Identification (G1 and G2)

For morphological identification, 20 adult specimens of each species (10 males and 10 females), unless otherwise described, were clarified in 80% acetic acid solution and diaphanized in vegetable creosote if necessary. The specimens were then mounted on temporary slides for observation on a microscope (Olympus BX-51) equipped with an Olympus BX-51 QColor3 digital camera (Olympus America Inc., Center Valley, PA, USA). Photomicrographs were obtained and processed using the Image-Pro Plus software (Media Cybernetics Inc., Bethesda, MD, USA). The morphometric data obtained were given in millimeters and composed of mean ± standard deviation. The morphological identification was based on the taxonomic keys proposed by Yamaguti [53], Vicente et al. [54], and Anderson et al. [55], in addition to the original species descriptions, when necessary. Vouchers of each species were deposited in LabEPar’s helminthological collection.

#### 4.4.2. Coprological and Urinary Analyses (G1)

At the laboratory, 22 stool and three urine samples were gathered from the colon and urinary bladder, respectively. The samples were identified and refrigerated until further analysis. Coprological analyses were performed using flotation and sedimentation techniques [56,57]. Egg counts (eggs per gram feces; EPG) were performed using the McMaster chamber [58]. Drops of urine were mounted on glass slides and observed directly under a light microscope. The morphometric data of eggs and oocysts were obtained and given as mentioned in Section 4.4.1.

### 4.5. Epidemiological Analyses (G1)

The standard descriptors of infection (prevalence, abundance, mean intensity, and range of intensity) were calculated [59]. To evaluate the effect of the host sex on these descriptors, data distribution was evaluated using Kolmogorov–Smirnov or Shapiro–Wilk tests. The effect of host sex on prevalence of infection was performed using Fisher’s exact test. The influence of host sex on parasite mean intensity was analyzed using the *t*-test or the Mann–Whitney U-test. The body condition index (BCI) of the hosts was determined by the body mass (kg) to total body length (cm) ratio. The association between the BCI and total parasite intensity was evaluated using Spearman’s correlation coefficient. All tests were performed using GraphPadPrism 5.0 software, with a *p*-value of 0.05. 

### 4.6. Molecular Analysis (G2)

#### 4.6.1. DNA Extraction, Amplification and Sequencing

Genomic DNA was extracted from an adult male specimen of each species, except for *Strongyloides ransomi*, for which one parthenogenetic female was used, and *Macracanthorhynchus hirudinaceus*, because only females were found. The specimens selected for DNA extraction were individually transferred to microtubes and were firstly washed with sterile PBS 1X solution pH 7.4. DNA extraction was performed by using the DNeasy Blood & Tissue kit (Qiagen, Hilden, Germany) according to the manufacturer’s protocol. All the specimens obtained were submitted to the amplification of ITS, 18S e 28S rDNA regions using the set of primers presented in Appendix A [60,61,62,63,64]. The reactions were composed of 1× buffer (50 mM KCl, 200 mM TRIS-HCl, pH 8.4); 50 mM of MgCl_2_; 10 mM of dNTP’s; 0.5 U of Platinum Taq DNA polymerase (Invitrogen, Thermo-Fisher Scientific, Waltham, MA, USA); 5 pmol of each Forward and Reverse primer; genomic DNA and ultrapure water to complete a final volume of 20 μL. Amplifications were performed in a Nexus thermal cycler (Eppendorf, Hamburg, Germany) programmed to perform one cycle at 95 °C for 3 min and 35 cycles at 94 °C for 40 s; each primer’s annealing temperature as shown in Appendix A was kept for 30 s, and 72 °C for 50 s, followed by a final cycle at 72 °C for 10 min. To verify amplification, the PCR products were submitted to electrophoresis in 1% agarose gel. In case of small helminths and consequently low DNA yield, the reamplification was performed using the same protocol and primers cited previously. When nonspecific bands were present, the band of interest was purified by excision of the agarose gel and purified with the Wizard® SV Gel and PCR Clean-Up System kit (Promega, Madison, WI, USA) according to the manufacturer’s instructions. The PCR products that presented a single band were purified directly from the microtube using the same kit. The purified PCR products were submitted to PCR sequencing using the BigDye Terminator v3.1 kit (Applied Biosystems, Waltham, MA, USA), according to the manufacturer’s instructions. Sequencing was performed by capillary electrophoresis on an ABI3130 sequencer (Applied Biosystems) [65].

#### 4.6.2. Phylogenetic Analysis

The electropherograms generated in the sequencing were submitted to the Phred/Phrap/Consed software package [66,67,68] to verify the quality of the bases and trim the sequences considering bases with Phred quality up to 20 or higher. The qualified sequences were compared to others deposited in the NCBI (National Center for Biotechnology Information) database using the BLAST tool [69]. The sequences from this study and the selected sequences from NCBI’s database (Appendix A) were then aligned using the MUSCLE tool [70]. For phylogenetic analyses, the best evolutionary model was selected according to Akaike’s information criterion (AIC) using the ModelTest 3.7 software [71,72]. Phylogenetic trees were obtained by Bayesian analysis using the Markov Chain Monte Carlo (MCMC) algorithm with MrBayes 3.2.3 software [73]. The evolutionary model used was substitution rate 6 and gamma distribution. The analyses were performed on four chains with 5,000,000 generations, and the trees were sampled every 100 generations. A standard deviation of less than 0.01 was obtained and 25% of the trees initially generated were discarded as burn in. The phylograms were graphically edited on the Dendroscope 3 software [74]. The sequences were submitted to GenBank, and their accession numbers can be found on Table 1. 

## 5. Conclusions

The results obtained provide evidence that wild boars retained some of their typical parasites common to suidae and have not acted as capture hosts for any species native to the Neotropical region. They may act as dispersers of these parasites to domestic swine, especially extensive low-tech subsistence farming, and possibly to native Tayassuida. The nematodes *G. urosubulatus*, *M. salmi*, *S. dentatus*, and *S. ransomi* are predominant in the wild boar parasitic community in the northeastern region of the state of São Paulo. Despite the pathogenic potential of the parasites identified, the animals evaluated had a good body score. The wild boar population studied showed no influence of sex on parasite infection indicators. 

In general, the morphological and morphometric studies of the species analyzed were outdated and incomplete, especially for *S. dentatus*. This work brought detailed morphological descriptions and standardized morphometric data. Even with the advance of molecular techniques for helminth species identification, knowledge of morphology is still essential and both techniques should be used in a complementary way according to the principles of integrative taxonomy.

It was possible to notice a lack of molecular studies for almost all the species analyzed. The sequences obtained in this study will significantly contribute to future works, and it is important that molecular studies continue to be expanded in the helminthology field. A good molecular database will make it possible to carry out work using non-invasive samples such as feces, for example, purely morphological studies of such samples, especially from wild animals, are very limited in helminthology.

## Figures and Tables

**Figure 1 pathogens-12-00175-f001:**
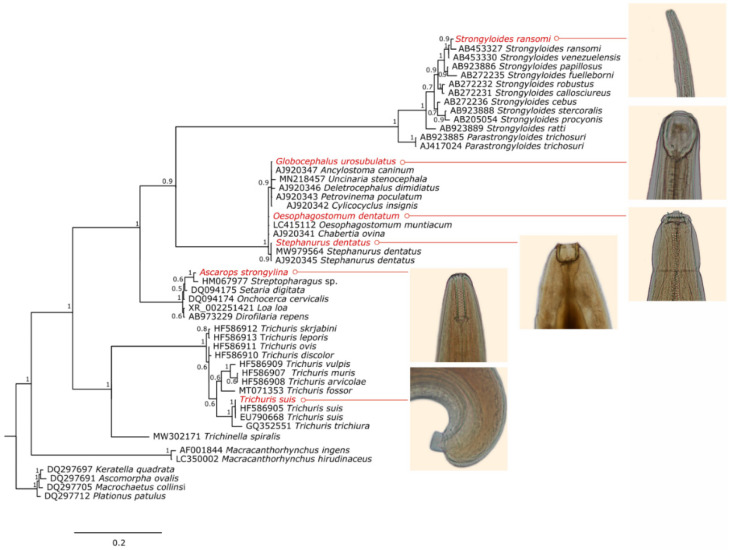
Bayesian phylogenetic tree of the 18S rDNA region encompassing parasites of the phyla Nematoda and Acanthocephala, with Rotifers as an outgroup. The sequences from this study are highlighted in red.

**Figure 2 pathogens-12-00175-f002:**
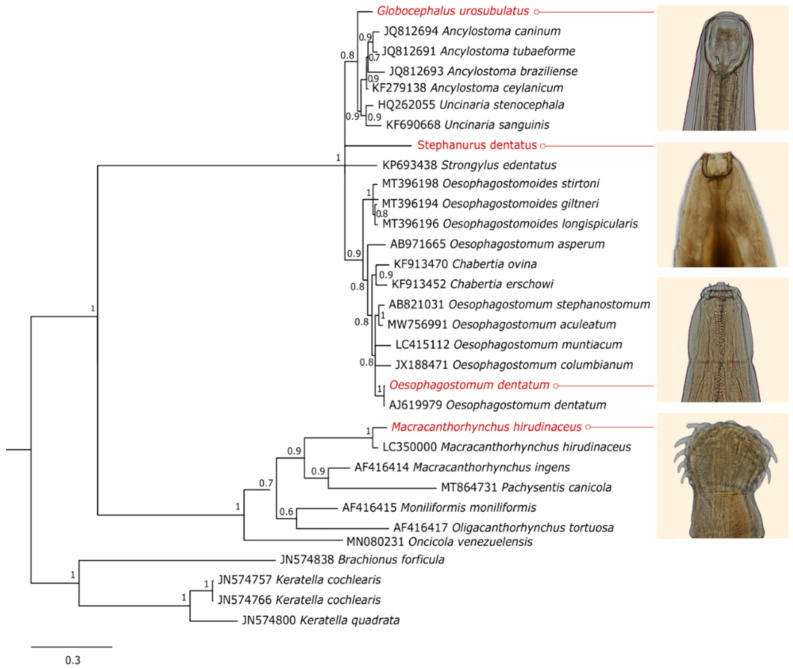
Bayesian phylogenetic tree of the ITS rDNA region encompassing parasites of the phyla Nematoda and Acanthocephala with Rotifers as an outgroup. The sequences from this study are highlighted in red.

**Figure 3 pathogens-12-00175-f003:**
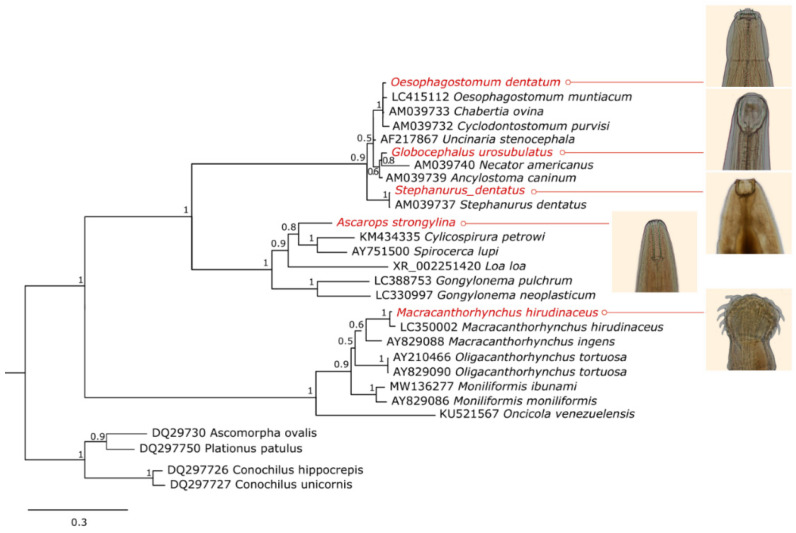
Bayesian phylogenetic tree of the 28S rDNA region encompassing parasites of the phyla Nematoda and Acanthocephala, with Rotifers as an outgroup. The sequences from this study are highlighted in red.

**Figure 4 pathogens-12-00175-f004:**
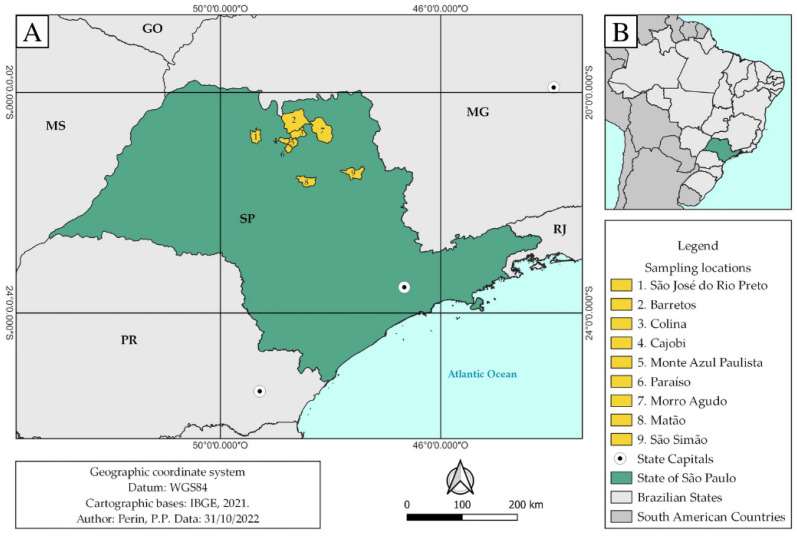
(**A**) Map of the state of São Paulo highlighting the northeastern cities in which wild boars were killed by wildlife controllers for this study: Barretos, Cajobi, Colina, Matão, Monte Azul Paulista, Morro Agudo, Paraíso, São José do Rio Preto, and São Simão. (**B**) Map of South America highlighting the location of Brazil and the state of São Paulo.

**Table 1 pathogens-12-00175-t001:** Accession numbers of the 18S, 28S, and ITS ribosomal regions amplified from each helminth species from wild boars hunted in São Paulo state, Brazil.

Species	18S rDNA	28S rDNA	ITS Region
*Ascarops strongylina*	OP288106	OP289657	-
*Globocephalus urosubulatus*	OP288108	OP289658	OP289650
*Strongyloides ransomi*	OP288111	-	-
*Oesophagostomum dentatum*	OP288109	OP289659	OP289651
*Trichuris suis*	OP288107	-	-
*Stephanurus dentatus*	OP288110	OP289661	OP289653
*Macracanthorhynchus hirudinaceus*	-	OP289660	OP289652

## Data Availability

Not applicable.

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
