# Peer review of "Epidemiology and Integrative Taxonomy of Helminths of Invasive Wild Boars, Brazil"

_pathogens, 2023, doi:10.3390/pathogens12020175_

Round 1

Reviewer 1 Report

The article , "Epidemiology and integrative taxonomy of invasive helminths of wild boars, Brazil" is a very interesting publication on the occurrence of worms in wild boars in Brazil.
From an epidemiological point of view, wild boars are excellent vectors of various parasites, giving us a lot of opportunities for their detection and research on their taxonomy.
The publication was prepared in a thorough and careful manner. The text was written in the third person, thank you.
The layout of the work is correct.
The methods of detection and molecular characterization used are comprehensively and thoroughly described, which can be found in the additional materials too.
From my point of view, this research work brings a lot of valuable information on the epidemiology and detection of helminths in wild boars found in Brazil.

Author Response

We would like to thank you for your revision and considerations.

Reviewer 2 Report

Title:

The title indicates the two major objectives of the study, Epidemiology and integrative taxonomy.

Epidemiological data is almost missing or not presented very well. 

G2 data did not use for epidemiological analyses. Why?

The association of age was not described in the results. 

Abstract

Results should be presented with statistical values. 

Some of the research findings are missing in the abstract.

Introduction

Establish the context by providing a brief and balanced review of the relevant published literature.

Material and methods:

Although some techniques (like DNA extraction and the McMaster technique) have a standard protocol, it is better to describe them briefly in each paper.

Figure 4: Labelling of Map A and Map B is missing.

Heading 4.4.2: why only 22 Coprological and three urinary samples were analysed?

Results:

Statistical values are missing in the results, especially in epidemiological analyses.

The results of different measurements of examined species should be in tabulated form like:

Character      Species 1          Species 2      so on

Total length 

Buccal capsule 

Esophagus 

Discussion

Authors should relate their work to the findings of other studies - including previous studies you may have done and those of other investigators

Line 15: ITS regions (write it completely)

Line 19: strongylina,and (space issue)

some of the basic comments are given in the attached file

Author Response

We would like to thank you for your revision and considerations. All corrections to the manuscript based on your suggestions are highlighted in yellow. 

1. Title
1.1 The title indicates the two major objectives of the study, Epidemiology and integrative taxonomy. Epidemiological data is almost missing or not presented very well. 
There are several limitations in sample acquisition of wild animals, which hinders the procurement of more robust epidemiological data. Nevertheless, the epidemiological and ecological data obtained are still useful and important. We added information to headings 2.1 and 4.5 to improve the presentation of the epidemiological data.

1.2 G2 data did not use for epidemiological analyses. Why?
This research paper compiles data from two different graduate researchers. One of them focused on the epidemiological and ecological aspects of helminth infection of the digestive and urinary tracts of wild boars. The second one focused on an integrative taxonomical analysis of those helminths. The samples for each study were gathered at different time periods and with different standardizations, since the research goals were not the same.

1.3 The association of age was not described in the results. 
Our samples have an age bias, since most of the animals hunted were piglets. Therefore, we didn’t analyze the association of age with helminth infection. We added this information to the results (item 2.1), for clarity.

2. Abstract: Results should be presented with statistical values. Some of the research findings are missing in the abstract.
We included statistical values and the missing research findings to the abstract.

3. Introduction: Establish the context by providing a brief and balanced review of the relevant published literature.
We believe that our introduction already stated the problem, and we would like to keep it as is.

4. Material and methods: Although some techniques (like DNA extraction and the McMaster technique) have a standard protocol, it is better to describe them briefly in each paper.
We kindly disagree and consider that citing the standard protocol methods is sufficient.

5. Figure 4: Labelling of Map A and Map B is missing.
We labelled the figure.

6. Heading 4.4.2: why only 22 Coprological and three urinary samples were analysed?
The animals frequently evacuate and urinate during the hunting, hindering the acquisition of fecal and urinary samples.

7. Results: Statistical values are missing in the results, especially in epidemiological analyses.
We added information to heading 2.1 to include the missing data.

6. The results of different measurements of examined species should be in tabulated form like:
Character       Species 1          Species 2      so on
Total length 
Buccal capsule 
Esophagus 
If we group the results of different measurements of all the seven species on the same table, we won’t be able to compare our data to different studies like we are currently doing. It would also result in a very long table. Therefore, we would like to keep them as is.

7. Discussion: Authors should relate their work to the findings of other studies - including previous studies you may have done and those of other investigators
We kindly disagree, as we related our work to all helminth diversity studies in Brazil and to the most recent and detailed works on wild boar and pig related helminths worldwide, as listed below:

Mundim, M.J.S.; Mundim, A.V.; Santos, A.L.Q. et al. Helmintos e protozoários em fezes de javalis (Sus scrofa scrofa) criados em cativeiro. Arq Bras Med Vet e Zootec 2004, 56, 792–795. https://doi.org/10.1590/S0102-09352004000600015 (In Portuguese)

Gomes, R.A.; Bonuti, M.R.; Almeida, K. de S.; Nascimento, A.A. do. Infecções por helmintos em Javalis (Sus scrofa scrofa) criados em cativeiro na região Noroeste do Estado de São Paulo, Brasil. Ciência Rural 2005, 35, 625–628. https://doi.org/10.1590/s0103-84782005000300021 (In Portuguese)

Da Silva, D.; Müller, G. Parasites of the respiratory tract of Sus scrofa scrofa (wild boar) from commercial breeder in southern Brazil and its relationship with Ascaris suum. Parasitol Res 2013, 112, 1353–1356. https://doi.org/10.1007/s00436-012-3214-1Da

Silva, D.; Müller, G. Parasitic helminths of the digestive system of wild boars bred in captivity. Rev Bras Parasitol Vet 2013, 22, 433–436.

Marques, S.M.T.; Sato, J.P.H.; Barcellos, D.E.S.N. Parasitos intestinais de javalis (Sus scrofa) criados na região sul do brasil. Ars Vet 2016, 32, 31–34. https://doi.org/10.15361/2175-0106.2016v32n1p31-34 (In Portuguese)

Dakova, V.; Panayotova-Pencheva, M. Morphometric Features of Oesophagostomum dentatum, O. quadrispinulatum and Ascarops strongylina in Materials from Wild Boars from Bulgaria. Acta morphol anthropol 201724, 3-4.

Giang, N.T.; Hoan, T.D.; Huyen, N.T.T. et al. Morphological and molecular characterisation of Strongyloides ransomi (Nematoda: Strongyloididae) collected from domestic pigs in Bac Giang province, Vietnam. Tap chi Sinh hoc 201739, 270-275. https://doi.org/10.15625/0866-7160/v39n3.9829

Pinheiro, R.H.D.S.; Melo, S.; Benigno, R.N.M.; Giese, E.G. Globocephalus urosubulatus (Alessandrini, 1909)(Nematoda: Ancylostomatidae) in Brazil: a morphological revisitation. Rev Bras de Parasitol Vet 2021, 30. https://doi.org/10.1590/S1984-29612021078

Lisitsyna, O.I. Fauna of Ukraine, Vol. 31. Acanthocephala; Naukova Dumka: Kyiv, Ukraine, 2019; pp. 1–223. (In Russian)

Amin, O.M.; Heckmann, R.A.; Dallarés, S. et al. New morphological and molecular perspectives about Macracanthorhynchus hirudinaceus (Acanthocephala: Oligacanthorhynchidae) from wild boar, Sus scrofa Linn., in Ukraine. J Helminthol 2021, 95. https://doi.org/10.1017/S0022149X21000675

Taylor, M.A; Coop R.L.; Wall R.L. Veterinary parasitology, 4th ed.; John Wiley & Sons: New Jersey, USA, 2015; pp 316-352.

Souza, H.C. Helmintos intestinais de Tayassuidae e Suidae (Mammalia: Artiodactyla) no Pantanal: um estudo sobre a circulação de espécies na Reserva Particular do Patrimônio Nacional SESC Pantanal e seu entorno, Barão de Melgaço, Mato Grosso, Brasil. Master dissertation, Fundação Oswaldo Cruz, Rio de Janeiro, 2014. (In Portuguese)

Mansouri, M.; Sarkari, B.; Mowlavi, G.R. Helminth parasites of wild boars, Sus scrofa, in Bushehr Province, Southwestern Iran. Iran J of Parasitol 201611, 377.

Panayotova-Pencheva, M.; Dakova, V. Studies on the gastrointestinal and lung parasite fauna of wild boars (Sus scrofa scrofa L.) from Bulgaria. Annals of Parasitology 201864. https://doi.org/10.17420/ap6404.174

Okoro, C.K.; Wilson, B.S.; Lorenzo-Morales, J.; Robinson, R.D. Gastrointestinal helminths of wild hogs and their potential livestock and public health significance in Jamaica. J Helminthol 201690, 139-143. https://doi.org/10.1017/S0022149X14000881

Daubney, R. The kidney-worm of Swine: a short Redescription of Stephanurus dentatus Diesing, 1839. J Comp Pathol Ther 192336, 97-103.

Neveu-Lemaire, M. Traité d'helminthologie médicale et vétérinaire, 1st ed.; Vigot: Paris, France, 2013; pp. 1-19. (In French)

8. Line 15: ITS regions (write it completely)
Done.

9. Line 19: strongylina,and (space issue)
Done.

10. Basic comments from the attached file:

10.1 Keywords: Wild Boar and Invasive Species should start with capital letter
Done.

10.2 Lines 391 and 392
We changed "nº" to "#". We meant "number".

10.3 Line 452 Use the same format throughout the text, somewhere a space is present and somewhere it is absent
We changed it to keep the same format along the text.

10.4 Line 495 There should be a scape between number and unit correct throughout the text
Done.

10.5 Correct references 667-669, 707, 734
Done.

Round 2

Reviewer 2 Report

Accepted